# Characteristics of 2D Ultrasonic Vibration Incremental Forming of a 1060 Aluminum Alloy Sheet

**DOI:** 10.3390/ma17061235

**Published:** 2024-03-07

**Authors:** Yuan Lv, Yifan Wang, Yan Wang, Xixiang Pan, Cong Yi, Meng’en Dong

**Affiliations:** College of Mechanical Engineering, Xi’an University of Science and Technology, Xi’an 710054, China; wyf19106591840@outlook.com (Y.W.); wangyan076@outlook.com (Y.W.); pan2504335@outlook.com (X.P.); 22205224086@stu.xust.edu.cn (C.Y.); 21205016010@stu.xust.edu.cn (M.D.)

**Keywords:** aluminum alloy, incremental forming, ultrasonic vibration, plasticity, micro-morphology, residual stress

## Abstract

Currently, 1060 aluminum alloy is widely applied in the electronics industry, construction, the aerospace field, traffic engineering, decorations, and the consumer goods market for its good chemical, physical, and mechanical properties. In general, excellent processing property is necessary and important for the manufacturing of complicated panels. In this paper, a special 2D ultrasonic vibration incremental forming method is designed to improve its plasticity and mechanical properties. Three kind of processing methods, including traditional single-point incremental forming, longitudinal ultrasonic vibration incremental forming, and 2D ultrasonic vibration incremental forming, are used for the flexible manufacturing of cones and cylindrical cups of 1060 aluminum alloy sheet. Then, micro-hardness tests, residual stress tests, and scanning electron microscopy tests are carried out to probe the changes in micro-structure and mechanical properties and to analyze the effects of different types of ultrasonic vibration on the plasticity and fracture characteristic of 1060 aluminum alloy. It is proven that 2D ultrasonic vibration facilitates the improvement of plasticity and surface qualities of 1060 aluminum alloy better than the other two processing methods. Therefore, the novel 2D ultrasonic vibration incremental forming process possesses substantial application value for the flexible and rapid manufacturing of complicated thin-walled component of aluminum alloy.

## 1. Introduction

Currently, 1060 aluminum alloy is widely applied in the electronics industry, construction, the aerospace field, traffic engineering, decorations, and the consumer goods market due to its excellent chemical, physical, and mechanical properties, such as high ductility, electrical conductivity, thermal conductivity, and corrosion resistance [1]. It is not suitable for heat treatment strengthening, soldering, and mechanical cutting. Plastic processing technology, like punching, wire drawing, extrusion, or rolling, is one of the most frequently used method for the manufacturing of metal products. The most common products include but are not limited to the following items: capacitors, gaskets, isolation nets of vacuum tubes, protective covers for electrical cables, components of ventilation systems of aircrafts, heat-conducting elements, radiators, hull parts of ships, metal handicrafts, doors, and window decorations [2,3]. Obviously, there are huge market requirements for thin-walled components of 1060 aluminum alloy and their manufacturing technology.

Punching is a kind of plastic forming technology, including deep drawing, bending, spinning, hydraulic bulging, creep age forming, blanking, and flanging, in which a metal sheet is deformed or detached under the impact force exerted by a die [4,5,6]. It is well known that the die-based manufacturing mode can meet the high efficiency and contour consistency demands of the mass production of single-variety products. However, it is not suitable for small-batch production of multiple-variety products due to its long manufacturing cycle and the high manufacturing costs of die. One kind of incremental forming technology is designed to satisfy the flexible manufacturing need for multiple-variety products [7,8,9,10,11,12,13]. With this die-free manufacturing technology, the metal sheet is extruded by a hemisphere tooling that feeds according to the contour lines of the target product instead of a rigid die with a molded surface. Convenient processing path programming could rapidly respond to the contour variations of the target products. Therefore, this flexible manufacturing technology has advantages in customized and rapid manufacturing, as well as being low cost and improving the technological properties of difficult-to-process metal materials.

In recent years, considerable research attention has been focused on high-energy assisted plastic processing of metal materials. Many new manufacturing methods, such as ultrasonic cutting, ultrasonic laser processing, and ultrasonic rolling, have been designed, and the mechanical behaviors and micro-structure changes of metal materials under multiple energy field conditions have been researched [14]. Khan [15] suggested a heat treatment and incremental forming processing route to produce age-hardened components with reasonable accuracy and formability. In addition, many studies have suggested that ultrasonic vibration could decrease forming force, soften materials, and improve contact friction conditions, which are beneficial for the improvement of manufacturing qualities. Considerable attention has also been focused on ultrasonic-assisted plastic forming technology. In Thanh’s paper, the influence of ultrasonic vibration on the forming forces and surface quality of products was investigated and compared with single-point incremental forming [16]. Yan Li [17] studied the influence of different vibration parameters on the formability of 1060 aluminum alloy sheet. In Liu Shen’s paper [18], ultrasonic vibration is imposed on the die to decrease the difficulties of drawing deformation of titanium wire at room temperature. The effect of longitudinal amplitude and frequency of ultrasonic vibration on deformation force was investigated. Research has also indicated that longitudinal and torsional composite vibration is effective for the improvement of surface quality. In Gohil’s research work, the effect of amplitude of vibration along with forming feed and step depth were investigated by conducting a groove test on AA3003 aluminum sheet [19]. Zhou Haiyang [20] studied the influence of frequencies and amplitudes of ultrasonic vibration on the plasticity of aluminum and titanium. To date, the effects of ultrasonic vibration parameters, including frequency and amplitude, on the mechanical behaviors and forming qualities of metal sheet have been investigated by several researchers, and some research achievements have been published in the literature [21,22,23,24]. However, few studies on the influence of strengthening ultrasonic energy and multi-dimensional mode of vibration on the mechanical performance and forming mechanism of metal materials have been reported. Therefore, it is necessary to design a new enhanced ultrasonic vibration incremental forming technology and discover the softening action of high ultrasonic energy on metals.

For enhancement of the volume effect and surface effect induced by ultrasonic energy, one kind of 2D ultrasonic vibration incremental forming method is used for the manufacturing of cones and cylindrical cups of 1060 aluminum alloy to study its plastic deformation behaviors and fracture characteristic. A novel 2D ultrasonic vibration incremental forming experiment for cones and cylindrical cups of 1060 aluminum alloy and contrast experiments are performed in this paper.

## 2. Materials and Methods

### 2.1. Experimental Materials

For its good usability and extensive applications in industries, 1060 aluminum alloy sheets (produced by the Alnan company of China, Nanning, China) are used for the experimental research using manufacturing tests, performance measurements, and mechanism analysis of cones and cylindrical cups in this paper. Its chemical composition is listed in Table 1.

### 2.2. Specimens and Tools

Incremental forming is one kind of variant stamping technology for the manufacturing of thin-walled components made of metal materials. The rolling blank is cut to be a semi-finished specimen with a three-dimensional size of 80 mm × 80 mm × 2 mm and 80 mm × 80 mm × 1 mm on the wire electrical discharge machine. Specimens of 1060 sheet and the die are shown in Figure 1a. Although this forming process is a die-free manufacturing method for thin-walled plates, an ordinary die playing a supporting role can facilitate the improvement of forming qualities. The die undergoing impact is made of 45 steel with good rigidity and high strength. The combined die is divided into three parts, including the fixture part, the core die part, and the blank holder. The core die with a replaceable surface is fixed on the vibrated platform by the fixture part. The specimen is put on the die and compressed by the blank holder and four bolts with preload. Strong pressing force and friction can prevent the free flow of the materials in the flange area of metal sheet. Therefore, the very common phenomenon of wrinkling occurring in the stamping process is avoid. The installation method is shown in Figure 1b. Because it undergoes coupling action with high temperature, violent striking, and intense friction, the hemispherical forming tool is made of a high-temperature alloy, K25 (produced by Baosteel company, Shanghai, China), as shown in Figure 1c.

### 2.3. Method of Experiment

#### 2.3.1. Processing Principle

In this paper, a kind of two-dimension ultrasonic vibration device is applied to the manufacturing of cones and cylindrical cups of 1060 aluminum alloy. As shown in Figure 2a, traditional single-point incremental forming (SPIF) is a variant stamping of layered extrusion that utilizes the motion path of a rolling forming tool instead of a traditional stamping die. As shown in Figure 2b, the existing longitudinal ultrasonic vibration incremental forming (LUVIF) method inherits the flexible forming merits of SPIF technology. It also decreases the forming force and improves the surface qualities by virtue of the volume effect and surface effect. Unlike the former two technologies, the new two-dimensional ultrasonic vibration incremental forming (2D-UVIF) method shown in Figure 2c consists of two ultrasonic vibration systems. On the one hand, the two-vibration-system design, which provides strengthening ultrasonic energy for softening difficult machining materials and decreasing the forming force, facilitates the manufacturing of complicated thin-walled components of high-strength alloys. On the other hand, its unique two-dimensional ultrasonic vibration motion is designed for increasing the contact areas between the metal part and the forming tool and improving the bad friction conditions between them. This is helpful for the improvement of surface qualities and surface conditions, including surface morphology, mechanical properties, and stress state. It is known that good surface qualities and excellent surface conditions can enhance mechanical properties and fatigue resistance performance.

#### 2.3.2. Processing Equipment

To complete the manufacturing tests for cones and cylindrical cups of 1060 aluminum alloy made with 2D ultrasonic vibration incremental forming, a new 2D ultrasonic vibration auxiliary device is designed and manufactured. It includes an h-shaped base, one X-axis ultrasonic vibration system, one Z-axis ultrasonic vibration system, and a vibrator platform. Each ultrasonic vibration system contains a signal generator, a signal amplifier, and an ultrasonic horn. The signal generator launches an ultrasonic wave signal that is amplified by the amplifier. Then, the strengthening signal drives the horn to vibrate at a frequency of 20 kHz and amplitude of 10 μm. As shown in Figure 3, in the two crossed ultrasonic vibration systems, the vibrating direction of one system is parallel to the axis of the forming tool, the vibrating direction of the other one is perpendicular to the axis of the forming tool, and both are connected to the platform. The part is made to vibrate in a circular track relative to the forming tool, while the two ultrasonic vibration devices vibrate at the same frequency and amplitude. The phase difference of the two signal waves is one quarter.

In addition, the vibrating track of the compound ultrasonic vibration device is customizable according to processing requirements. For example, its vibrating track is an ellipse if the two ultrasonic vibration devices vibrate at the same frequency and different amplitudes, and their phase difference is a quarter. In addition, non-vibration mode, one-way horizontal vibration mode, one-way vertical vibration mode, and compound diagonal vibration mode are also realizable.

As shown in Figure 3b, the 2D ultrasonic vibration auxiliary device (produced by Haituo Machinery Technology company, Handan, China) is fixed on the workbench of a four-axis CNC machining center (produced by Jinan First Machine Tool company, Jinan, China). A fixture and a conical-surface die are installed on the vibrating platform of the 2D ultrasonic device. Additionally, a hemispherical tooling is designed for replacing the traditional milling tooling. Then, the universal milling machine is transformed to be a specialized 2D ultrasonic vibration incremental forming machine.

#### 2.3.3. Processing Conditions

The manufacturing experiments on cones and cylindrical cups of 1060 aluminum alloy made with traditional incremental forming, longitudinal vibration incremental forming, and 2D ultrasonic vibration incremental forming are performed. Firstly, blank sheets (80 mm × 80 mm × 2 mm) of 1060 aluminum alloy are installed on the cone die, and blank sheets (80 mm × 80 mm × 1 mm) of 1060 aluminum alloy are installed on the cylindrical cup die. Secondly, a processing procedure is developed and loaded in the operating system of the specialized incremental forming machine. Thirdly, tool setting is completed accurately, and then the workpiece coordinate system is established. Fourthly, processing parameters are set in the operating system. The hemispherical forming tool feeds horizontally at a constant velocity of 500 mm/min and feeds vertically at a layer depth of 0.1 mm, as well as rotating at a speed of 500 r/min. Fifthly, the cooling system and the ultrasonic vibration system are opened. Fully synthetic cutting fluid of S318 (produced by Ganis company, Shenzhen, China) is poured on the blank sheet of 1060 aluminum alloy and the forming tool of K25 to cool them down. Additionally, this cooling and lubricating fluid is also helpful for the improvement of friction conditions and surface qualities. Then, the traditional incremental forming, the longitudinal vibration incremental forming, and the 2D ultrasonic vibration incremental forming tests are performed automatically by the processing program. The cone surface and the cylindrical cup surface are split into many contour circle lines by a series of horizontal planes. The forming tool feeds horizontally along the first contour line for one round, and then feeds vertically along the inside wall of the die to the second contour line. It constantly reproduces the feeding movement until the processing procedure is finished. These continuous trajectories of the different kinds of incremental forming methods for cone and cylindrical cup are displayed in Figure 4a,b.

Three cones processed by traditional single-point incremental forming, longitudinal ultrasonic vibration incremental forming, and 2D ultrasonic vibration incremental forming are shown in Figure 5. For its good processing property, these three thick cups of the same size, without wrinkling or fracture problems, are mainly used for micro hardness tests and residual stress tests.

To discover the effect of the 2D ultrasonic vibration mode on the improvement of machined surface qualities and plastic deformation capacity, three 1060 aluminum alloy sheets of 80 mm × 80 mm × 1 mm are used for manufacturing tests on cylindrical cups made with traditional single-point incremental forming, longitudinal ultrasonic vibration incremental forming, and 2D ultrasonic vibration incremental forming. Because of extensive deformation occurring at the bottom of the side wall, it is likely to rupture at this location. Once the rupture phenomenon is detected, the processing procedure will be terminated immediately. The machined surfaces are used for surface topography analysis with laser spectral confocal microscopy (LSCM of KC-H020, produced by Kathmatic company, Nanjing, China) and micro morphology analysis with a scanning electronic microscope (SEM of VEGA, produced by Tescan company, Brno, Czech). The fracture surfaces are probed by SEM technology. These fracture measurements are facilitated to research the micro-structure changes in 1060 aluminum alloy under coupled conditions of ultrasonic, temperature, and stress. Three cylindrical cups of different size are shown in Figure 6. The measurement locations are displayed in Figure 6. The results prove that the 2D ultrasonic vibration method is more helpful than the longitudinal ultrasonic vibration method for improveming the plastic deformation capacity of 1060 aluminum alloy. Compared to the cylindrical cup of SPIF, the depth of the 2D-UVIF cup is increased by 43.5%. However, the depth of the cylindrical cup of LUVIF is only raised by 17.7%. Obviously, the novel 2D-UVIF method has more advantages in enhancing the plasticity of 1060 aluminum alloy.

## 3. Results and Discussion

Three cones and three cylindrical cups are manufactured separately by using traditional single-point incremental forming, longitudinal ultrasonic vibration incremental forming, and 2D ultrasonic vibration incremental forming technology. To investigate the effect of the newly designed 2D ultrasonic vibration method on the mechanical behaviors and forming mechanism of 1060 aluminum alloy, surface topography tests, micro-hardness tests, residual stress tests, and scanning electron microscope observation tests of the machined surface and fracture surface of cones and cylindrical cups are conducted.

### 3.1. Surface Topography Analysis

To investigate the characteristics of the machined surface finish of cylindrical cups processed by different forming technologies, a set of surface topography tests are conducted. Sampling locations are displayed in Figure 6. Three surface topography images are shown in Figure 7; in these images, the red color represents peak height, and the blue color represents valley depth. It is found that there are similar ranges of maximum peak height and maximum valley depth in the three images. However, totally different topography features are distributed on the machined surfaces of the cylindrical cups. 

To quantitatively analyze the topography characteristics and roughness of the machined surfaces of the three cylindrical cups processed by the SPIF, LUVIF, and 2D-UVIF methods, the measurement results of the surface topography parameters, including mean roughness (Ra), maximum peak height (Rp), root mean square deviation (Rq), maximum valley depth (Rv), and maximum height difference between maximum peak height and maximum valley depth (Rz), are presented in Table 2, Table 3 and Table 4. As the three tables show, the mean roughness of the machined surfaces of the LUVIF cylindrical cup is larger than those of the other two. The 2D-UVIF technology produces a minimum mean roughness of the machined surfaces of the cylindrical cup. Moreover, the parameters of Rp, Rq, Rv, and Rz feature the same trends. Obviously, the novel 2D-UVIF method has good advantages with regard to machined surface qualities.

### 3.2. Micro-Hardness Analysis

It is well known that surface hardness is one of the most important mechanical properties of metal materials. The micro-hardness test is widely used to detect the mechanical properties of raw materials and semi-finished or finished products because this non-destructive testing approach is unlimited with regard to the shape and size of the detected object, and hardness is always closely associated with the strength and toughness of metal materials. For detecting the mechanical properties of 1060 aluminum alloy sheets used with traditional single-point incremental forming, longitudinal ultrasonic vibration incremental forming, and 2D ultrasonic vibration incremental forming, a series of micro-hardness tests are performed on the HAMS automatic Vickers micro-hardness testing system. A right square pyramid diamond indenter is pressed on the surface of the detected object with a testing pressure of 2.9 N and testing time of 15 s. The hardness value is calculated automatically by dividing the pressure by the area of the testing square cone indentation shown in the Figure 8b. As presented in Figure 8a, there are ten sampling points located equidistantly and equiangularly in a helix line on the inside wall of the cones processed by traditional single-point incremental forming, longitudinal ultrasonic vibration incremental forming, and 2D ultrasonic vibration incremental forming technology. Simultaneously, ten sampling points randomly distributed on the surface of the 1060 aluminum alloy sheet are detected. All the micro-hardness test values are shown in Figure 8c. The black square points represent the surface micro-hardness values of the unprocessed 1060 aluminum alloy sheet. The blue rhombus points represent the surface micro-hardness values of the cone made with 2D ultrasonic vibration incremental forming. The green circle points represent the surface micro-hardness values of the cone made with traditional single-point incremental forming. The red star points represent the surface micro-hardness values of the cone made with longitudinal ultrasonic vibration incremental forming. 

Despite the few data outliers existing on the four curves, it is clear that the influences of the different processing technologies on the surface mechanical properties of 1060 aluminum alloy are totally distinct. As Figure 6c shows, the surface hardness values of the unprocessed 1060 aluminum alloy sheet are at their minimum. Their mean value and variance are 39.85 HV and 1.37. These testing data demonstrate that the mechanical properties of 1060 aluminum alloy are poor, and the large-scale industrial process provides good consistency in processing quality and mechanical properties. The maximum values are seen in the red circle points representing the surface hardness of the cone made with longitudinal ultrasonic vibration incremental forming. Some studies in this field have indicated that ultrasonic incremental forming generally produces coupling actions of ultrasonic softening, ultrasonic hardening, stress superposition, heat softening, and work hardening on metal materials. Obviously, longitudinal ultrasonic vibration, on the one hand, provides a high-frequency impact on the surface of the aluminum alloy sheet. On the other hand, it creates softening action inside of the metal materials, which is conducive to increasing its plasticity. Their mean value and variance are 84.60 HV and 14.21. Testing results prove that hardening dominates on the surface of the cone because of the longitudinal impact and cold-formed process. The impact hardening actions are also heterogeneously distributed, so the red curve shows features of poor consistency. The green curve is lower than the red one. Its mean value and variance are 61.50 HV and 7.93. The main reason is that traditional single-point incremental forming only produces a work-hardening effect. The blue curve indicates that the 2D ultrasonic vibration creates a strengthening and softening effect on the surface of the aluminum alloy part. Ultrasonic softening, stress superposition, and heat softening counteract the ultrasonic hardening and work hardening. Some testing hardness values are even lower than those of the unprocessed 1060 aluminum alloy sheet, proving this opinion. The two-dimensional vibration exerted on the poor rigid sheet does not provide impact hardening on the surface of the processed part. Lastly, the lower main value of 42.82 HV and variance of 5.12 show that softening dominates on the surface of the 1060 alloy, and the 2D vibration creates a uniform mechanical effect on its processed surface.

### 3.3. Residual Stress Analysis

In order to probe the distribution and magnitude of the residual stress field on the processed surfaces of cones made with traditional single-point incremental forming, longitudinal ultrasonic vibration incremental forming, and 2D ultrasonic vibration incremental forming, three sets of residual stress tests are performed on an i-XRD residual stress analysis system (μ-X360, produced by Pulstec industrial company, Shizuoka, Japan). The measurement conditions are listed in Table 5.

There are five sampling points located equidistantly and equiangularly in a helix line on the inside wall of each processed component. The sampling points are shown in Figure 9a. These sampling locations are stamped with distinguishing marks and probed on an i-XRD residual stress system. The testing procedure is demonstrated in Figure 9b. The main advantage of this sampling method is that it obtains more extensive information on the characteristics, including different depth directions and circumference directions, of the processed surface of the cone by using multiple sampling points. 

Three sets of residual stress testing results are displayed in Figure 9c. In this image, numbers on the horizontal axis represent sampling point numbers, and numbers on the vertical axis represent residual stress value. A positive value indicates residual tensile stress, and a negative value indicates residual compressive stress. The three groups of 15 data points are marked with different colors. The black points, the blue points, and the red points represent, respectively, the residual stress value on the surface of the cones for traditional single-point incremental forming, longitudinal ultrasonic incremental forming, and 2D ultrasonic vibration incremental forming. It is indicated that the residual stress distribution on the surface of the three components are of distinct patterns. Firstly, the residual stress at locations number 5 and number 4, which are close to the top of the cone, is tensile. Secondly, the residual stress at position number 3, which is located in the middle of the cone, is compressive. It is deduced that the combined action of the die and the forming tool provides a strong impact on the processed surface. The ultrasonic energy also enhances this impact action. Therefore, a larger residual compressive stress is distributed on the surface of the cone for longitudinal ultrasonic incremental forming and 2D ultrasonic vibration incremental forming. Obviously, the longitudinal ultrasonic vibration produces more impact hardening than 2D ultrasonic vibration. Thirdly, the residual stress field at the bottom of the cone has fallen close to zero. It is inferred that the unmatched-size forming tool could not reach the bottom of the cone. Therefore, there is no impact hardening in this area. In brief, these three forming technologies only produce residual compressive stress in local areas of the processed surface.

### 3.4. Microstructure Images

It is believed that the surface quality of a processed part is not only related to its aesthetics, but also related to its fatigue life and structure safety. The machined surface micro-morphology of 1060 aluminum alloy sheet and cylindrical cups made with single-point incremental forming, longitudinal ultrasonic vibration incremental forming, and 2D ultrasonic vibration incremental forming are probed with scanning electron microscope (SEM) technology. The micro-structure images are shown in Figure 10. Firstly, Figure 10a–c show the micro-morphology of the unprocessed surface of a 1060 aluminum alloy sheet. Images at three different magnifications indicate that it features a smooth surface as well as some tiny and slight rolling indentations. Figure 10d–f show the surface micro-morphology of the cylindrical cup made with single-point incremental forming. A considerable number of distinct indentations appear on its surface. It is reasonable to assume that this damage is related to the rapidly rotating forming tool, which exerts strong force on the part. This intense extruding force produces huge deformation as well as friction damage. In general, it is well known that ultrasonic vibration induces a surface effect that could improve friction conditions between the forming tool and processed part. This opinion is proven by Figure 10g–i. There are many fewer friction marks appearing on its surface compared to Figure 10d–f. However, an obvious scratch is found in Figure 10i. It is deduced that some particles detach from the alloy sheet due to strong extrusion and fall into the processing area between the sheet and the forming tool. Then, these hard solid pieces scratche the processed surface under the pressure and friction of the rotating forming tool. The longitudinal vibration of the forming tool intensifies this phenomenon. In Figure 10j–l, a few inconspicuous scratches are observed on the cylindrical cup made with 2D ultrasonic vibration incremental forming. This is mainly due to the newly designed two-dimensional ultrasonic vibration. However, there are still a lot of detached particles adhered on its surface. On the one hand, the strengthening surface effect is conducive to improving the friction conditions between the part and the tool. On the other hand, its unique vibrating design is helpful for driving those particles away from the processing area. Therefore, it avoids such scratching behaviors caused by hard particles that are driven by the rapidly rotating forming tool. In short, it is proven that the 2D ultrasonic forming has more advantages in improving the surface qualities than the other two methods.

It is generally accepted that the micro-morphology characteristics of a fracture surface contain abundant information about mechanical behaviors, deformation, and fracture. As shown in Figure 11a–c, a few small and superficial dimples are found on the fracture surface of the cylindrical cup made with traditional single-point incremental forming. This ductile fracture feature indicates that 1060 aluminum alloy is of good plasticity. However, its limited plastic deformation capacity prevents the development of these dimples. The small dimples have no chance to grow and then break. These adjacent dimples finally form tearing ridges. In Figure 11d–f, some large-size dimples of 20 μm in diameter appear on the fracture surface of the cylindrical cup made with longitudinal ultrasonic vibration incremental forming. It is inferred that the volume effect induced by the ultrasonic energy greatly enhanced its plastic deformation capacity. Therefore, the micro-dimples grow and aggregate to become a large dimple. Meanwhile, some secondary phase particles detach from the fracture surface. In addition, an evident hardening layer is found on the processed surface. This phenomenon is consistent with the micro-hardness testing results. As plastic deformation increases, some short cracks are created on the fracture surface. In Figure 11g–i, there are many more isometric and huge dimples of 10–20 µm in diameter appearing on the fracture surface of the cylindrical cup made with 2D ultrasonic vibration incremental forming. This proves that the strengthening ultrasonic vibration design is beneficial for improving the plastic deformation property of 1060 aluminum alloy. When the forming tool exerts mechanical force and ultrasonic energy on the metal materials, plentiful micro-dimples are created, and the adjacent dimples grow to become one large dimple. Some small dimples hide around the large ones. At the same time, some secondary phase particles detach from their original locations. Because of its good ductile strength, there are only a few short cracks observed on the fracture surface.

## 4. Conclusions

In this paper, a kind of 2D ultrasonic vibration incremental forming is applied for the manufacturing of cones and cylindrical cups of 1060 aluminum alloy. Compared to traditional single-point incremental forming and longitudinal ultrasonic vibration incremental forming, this novel forming technology inherits the flexible manufacturing merits of incremental forming. It also possesses remarkable advantages in the improvement of processing performance and surface qualities. Testing results indicate that strengthening ultrasonic energy is helpful for greatly improving the plastic deformation capacity of 1060 aluminum alloy, and a two-dimension vibrating design could greatly ameliorate the friction conditions between the forming tool and the processed part. In conclusion, this research proves that the novel 2D ultrasonic vibration incremental forming method is suitable for the rapid manufacturing of complicated components of 1060 aluminum alloy.

## Figures and Tables

**Figure 1 materials-17-01235-f001:**
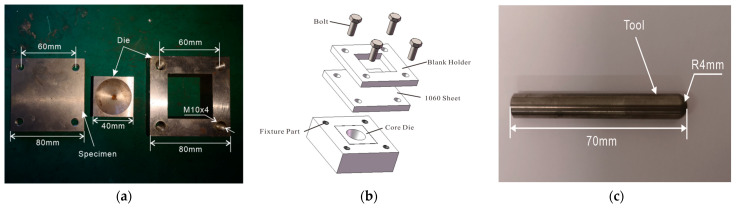
Devices for incremental forming: (**a**) die and specimen; (**b**) installation of die and specimen; (**c**) forming tool.

**Figure 2 materials-17-01235-f002:**
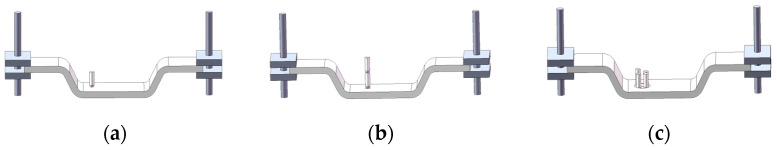
Processing principles of three forming technologies: (**a**) traditional single-point incremental forming (SPIF); (**b**) longitudinal ultrasonic vibration incremental forming (LUVIF); (**c**) 2D ultrasonic vibration incremental forming (2D-UVIF).

**Figure 3 materials-17-01235-f003:**
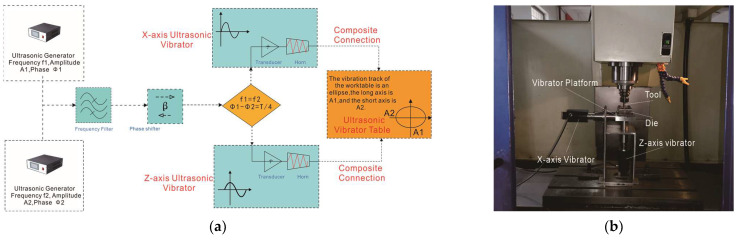
A specialized ultrasonic vibration incremental forming machine: (**a**) schematic design of 2D ultrasonic vibration system; (**b**) 2D ultrasonic vibration device.

**Figure 4 materials-17-01235-f004:**
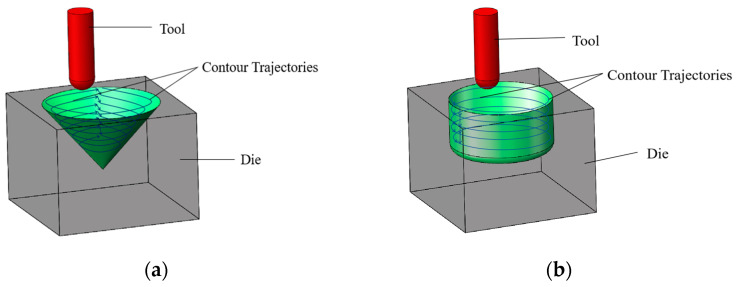
The continuous trajectories of the three incremental forming methods: (**a**) cone; (**b**) cylindrical cup.

**Figure 5 materials-17-01235-f005:**
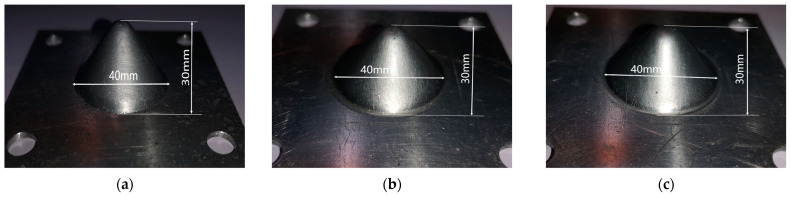
Processed cones: (**a**) SPIF; (**b**) LUVIF; (**c**) 2D-UVIF.

**Figure 6 materials-17-01235-f006:**
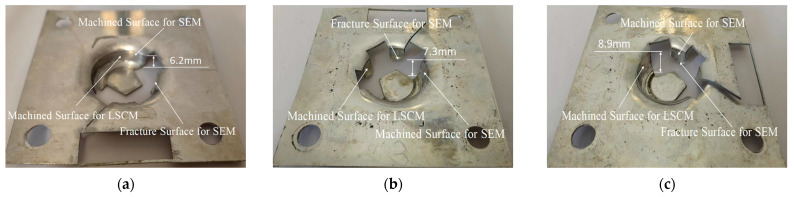
Processed cylindrical cups: (**a**) SPIF; (**b**) LUVIF; (**c**) 2D-UVIF.

**Figure 7 materials-17-01235-f007:**
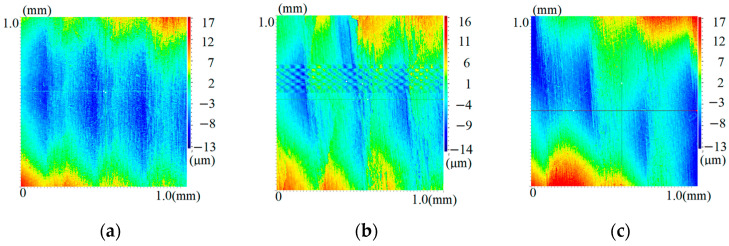
Machined surface topography images of cylindrical cups: (**a**) SPIF; (**b**) LUVIF; (**c**) 2D-UVIF.

**Figure 8 materials-17-01235-f008:**
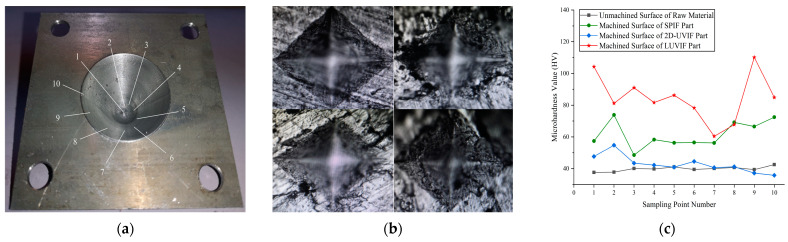
Micro-hardness tests: (**a**) sampling locations on the cone (ten testing points located equidistantly and equiangularly in a helix line on inside wall of cone); (**b**) micro-hardness testing images (images from the top left corner to the bottom right corner show the testing square cone indentation on the surface of the 1060 aluminum alloy sheet, SPIF part, LUVIF part, and 2D-UVIF part, respectively); (**c**) micro-hardness curves.

**Figure 9 materials-17-01235-f009:**
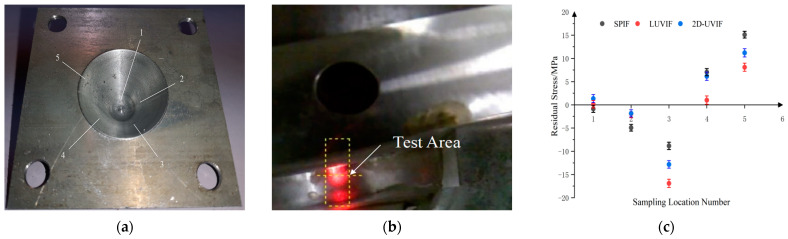
Sampling method and residual stress tests of cone: (**a**) sampling points on processed surface; (**b**) residual stress test procedure; (**c**) residual stress testing results.

**Figure 10 materials-17-01235-f010:**
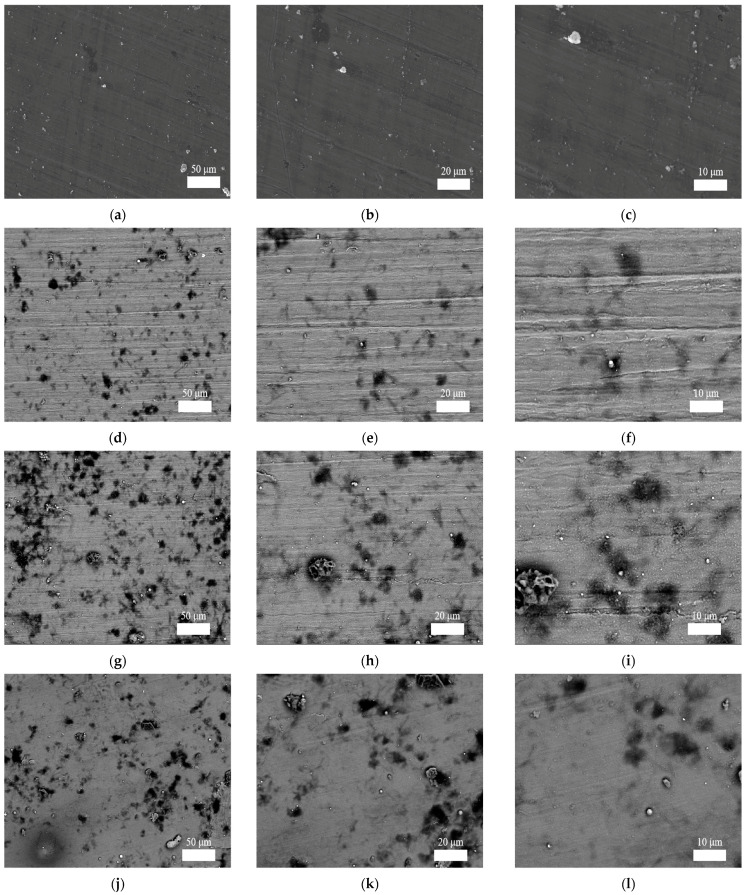
Magnification images of surface morphology of 1060 aluminum alloy sheet, SPIF part, LUVIF part, and 2D-UVIF part: (**a**–**c**) 1060 sheet; (**d**–**f**) SPIF; (**g**–**i**) LUVIF; (**j**–**l**) 2D-UVIF.

**Figure 11 materials-17-01235-f011:**
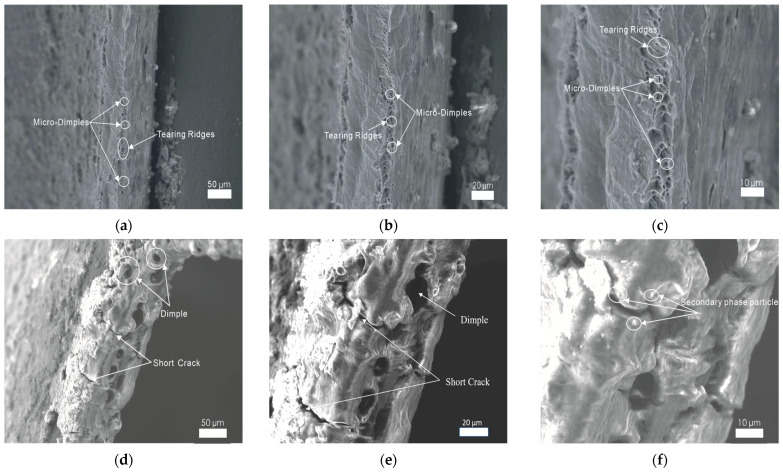
Micro-morphology images of fracture surface of the cylindrical cups for SPIF, LUVIF, and 2D-UVIF: (**a**–**c**) SPIF; (**d**–**f**), LUVIF; (**g**–**i**), 2D-UVIF.

**Table 1 materials-17-01235-t001:** Chemical composition of 1060 aluminum alloy (wt.%).

Si	Fe	Cu	Mn	Mg	V	Zn	Ti	Al
0.25	0.35	0.05	0.03	0.03	0.05	0.05	0.03	≥99.60

**Table 2 materials-17-01235-t002:** Measurement results of surface topography parameters of SPIF cylindrical cups (units: μm).

Topography Parameters	Max Value	Min Value	Mean Value	Standard Deviation
Mean roughness/Ra	3.150	1.691	2.421	1.032
Maximum peak height/Rp	9.679	5.425	7.552	3.008
Root Mean square deviation/Rq	3.657	2.097	2.877	1.103
Maximum valley depth/Rv	−5.136	−7.459	−6.298	1.643
Maximum height difference/Rz	17.138	10.562	13.850	4.650

**Table 3 materials-17-01235-t003:** Measurement results of surface topography parameters of LUVIF cylindrical cups (units: μm).

Topography Parameters	Max Value	Min Value	Mean Value	Standard Deviation
Mean roughness/Ra	3.077	1.888	2.661	0.670
Maximum peak Height/Rp	10.693	5.544	8.594	2.703
Root mean square deviation/Rq	4.031	2.311	3.346	0.912
Maximum valley depth/Rv	−5.617	−8.708	−7.590	1.713
Maximum height difference/Rz	19.401	11.162	16.184	4.406

**Table 4 materials-17-01235-t004:** Measurement results of surface topography parameters of 2D-UVIF cylindrical cups (units: μm).

Topography Parameters	Max Value	Min Value	Mean Value	Standard Deviation
Mean roughness/Ra	2.081	1.794	1.938	0.203
Maximum peak height/Rp	6.622	5.851	6.237	0.545
Root Mean square deviation/Rq	2.474	2.260	2.367	0.151
Maximum valley depth/Rv	−4.993	−7.004	−5.999	1.422
Maximum height difference/Rz	12.855	11.615	12.235	0.877

**Table 5 materials-17-01235-t005:** Measurement conditions for residual stress tests.

Parameters	Sample for SPIF	Sample for LUVIF	Sample for 2D-UVIF
Pitch	100 µm	100 µm	100 µm
X-ray irradiation measure time	90 s	90 s	90 s
X-ray irradiation max time	120 s	120 s	120 s
X-ray tube current	1.50 mA	1.50 mA	1.50 mA
X-ray tube voltage	30.00 kV	30.00 kV	30.00 kV
Sample distance (monitor)	65.000 mm	65.000 mm	65.000 mm
Sample distance (analysis)	63.908 mm	65.990 mm	66.174 mm
X-ray incidence angle	35.0°	35.0°	35.0°
Offset of alpha angle	0°	0°	0°
X-ray wavelength (K-alpha)	2.29093[A] (Cr)	2.29093[A] (Cr)	2.29093[A] (Cr)
X-ray wavelength (K-beta)	2.08480[A] (Cr)	2.08480[A] (Cr)	2.08480[A] (Cr)
Total measurement count	45,376	45,381	45,386
Oscillation count	7	7	7
Detection sensitivity	53.4%	39.6%	56.5%
Peak strength	317 k	144 k	211 k
Level of ambient light	0.3%	0.3%	0.3%
Temperature	38.38 °C	46.50 °C	43.69 °C
Valid range of alpha angle	18–90°	18–90°	18–90°
Peak analysis method	Fitting Lorentz	Fitting Lorentz	Fitting Lorentz

## Data Availability

All the data are available within the manuscript.

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
