# Peer review of "Characteristics of 2D Ultrasonic Vibration Incremental Forming of a 1060 Aluminum Alloy Sheet"

_materials, 2024, doi:10.3390/ma17061235_

Round 1

Reviewer 1 Report

Comments and Suggestions for Authors

The scientific work entitled "Effects of 2D Ultrasonic Vibration on the Plasticity and Fracture Characteristics of a 1060 Aluminum Alloy Sheet for Incremental Forming" deals with an interesting method of plastic forming of thin profiles using Incremental Forming. The contribution of 2D ultrasonic vibrations to the surface quality and internal stresses is evident. This work brings valuable information which inspiration for other research and industrial use. I have only minor suggestions for manuscript correction:

1. term "scholars" in the context is better to replace by "researchers" and "conical components" for "cones"

2. In table 1 description specify chemical composition is in wt.% or at.%?

3 Figure 6c unit MPa

4. "longitudinal axis" to "vertical axis"

Comments on the Quality of English Language

very good paper

Author Response

Dear Editors and reviewers,

On behalf of my co-authors, we thank you very much for giving us an opportunity to revise our manuscript, we appreciate editors and reviewers very much for their positive and constructive comments and suggestions on our manuscript entitled “Characteristics of 2D Ultrasonic Vibration Incremental Forming of a 1060 Aluminium Alloy Sheet”(ID: Materials-2874144).

We have studied reviewer’s comments carefully and have made revision which marked in red in the paper. We have tried our best to revise our manuscript according to the comments. The revised parts are list as follows.

Comment 1: “term "scholars" in the context is better to replace by "researchers" and "conical components" for "cones"” 

Answer: The manuscript has been revised.

“As so far, effects of ultrasonic vibration parameters, including frequency and amplitude, on mechanical behaviors and forming qualities of metal sheet have been investigated by several researchers and some research achievements have been published in the literature [21-24].”

“In this paper, one kind of 2 dimension ultrasonic vibration device is applied to manufacturing of cones and cylindrical cups of 1060 aluminium alloy.”

Comment 2: “ In table 1 description specify chemical composition is in wt.% or at.%?” 

Answer: The table 1 is revised.

Si

Fe

Cu

Mn

Mg

V

Zn

Ti

Al

0.25

0.35

0.05

0.03

0.03

0.05

0.05

0.03

≥99.60

Table 1. Chemical composition of 1060 aluminum alloy (wt.%)

Comment 3: “ Figure 6c unit MPa” 

Answer: The wrong unit in the figure is corrected.

Comment 4: “ "longitudinal axis" to "vertical axis"” 

Answer: It has been corrected. “Three sets of residual stress testing results are displayed in Fig.9(c). In this image, numbers on the horizontal axis represent sampling point numbers and numbers on the vertical axis represent residual stress value.”

In addition, other problems are also corrected. The revised manuscript is seen in the attachment.

We would like to express our great appreciation to you and reviewers for comments on our paper. Looking forward to hearing from you.

Thank you and best regards.

Yours sincerely,

Name: Yuan Lv

E-mail: lvyuan@xust.edu.cn

Mobile Phone: 86-13572199684

Reviewer 2 Report

Comments and Suggestions for Authors

Wire EDM is a process that creates a thermally influenced zone in the structure of the blank. At such a small blank size (80 x 80 mm) this thermally influenced zone calls into question all the results related to the structure of the resulting material after processing. Normally the cutting of blanks for SPIF processing is done on water jet cutting machines.

Figure 2 should show the moving parts and their trajectories (directions of movement). In the current version the figure does not make any scientific contribution. 

Nothing is specified about the type of trajectory used for SPIF pin processing (contour, spiral, etc). A graphical representation of these trajectories is needed. It is also not specified whether these trajectories are discontinuous (when moving from one Z coordinate to another the tool lifts off the blank) or continuous.

Author Response

Dear Editors and reviewers,

On behalf of my co-authors, we thank you very much for giving us an opportunity to revise our manuscript, we appreciate editors and reviewers very much for their positive and constructive comments and suggestions on our manuscript entitled “Characteristics of 2D Ultrasonic Vibration Incremental Forming of a 1060 Aluminium Alloy Sheet”(ID: Materials-2874144).

We have studied reviewer’s comments carefully and have made revision which marked in red in the paper. We have tried our best to revise our manuscript according to the comments. The revised parts are list as follows.

Comment 1: “Wire EDM is a process that creates a thermally influenced zone in the structure of the blank. At such a small blank size (80 x 80 mm) this thermally influenced zone calls into question all the results related to the structure of the resulting material after processing. Normally the cutting of blanks for SPIF processing is done on water jet cutting machines.” 

Answer: micro-structure tests are conducted on the fracture surfaces of the cylindrical cups by using SEM. The fracture surface is produced in the incremental forming process not by WEDM. WEDM is used for cutting the specimens from the cylindrical cups. The cutting locations are far away from the SEM testing locations. So, it will not affect its micro-structure.

“To discover the effect of the 2D ultrasonic vibration mode on the improvement of machined surface qualities and plastic deformation capacity, three 1060 aluminium alloy sheets of 80mm×80mm×1mm are used for manufacturing tests of cylindrical cups of traditional single point incremental forming, longitudinal ultrasonic vibration incremental forming and 2D ultrasonic vibration incremental forming. Because of extensive deformation occurred at the bottom of the side wall, it probably ruptures at this location. Once the rupture phenomenon is detected, processing procedure will be terminated immediately. Their machined surfaces are used for surface topography analysis of laser spectral confocal microscopy (LSCM of KC-H020, produced by Kathmatic company) and micro morphology analysis of scanning electronic microscope (SEM of VEGA, produced by Tescan company of Czech). And their fracture surfaces are probed by SEM technology. These fracture measurements are facilitated to research of micro-structure revolution of 1060 aluminium alloy under coupled conditions of ultrasonic, temperature and stress. Three cylindrical cups of different size are shown in figure 6. And the measurement locations are displayed in Fig.6. The results prove that 2D ultrasonic vibration method is more helpful than longitudinal ultrasonic vibration method for improvement of plastic deformation capacity of 1060 aluminium alloy. Compared to the cylindrical cup of SPIF, the depth of the one of 2D-UVIF is increased by 43.5%. However, the depth of the cylindrical cup of LUVIF is only raised by 17.7%. Obviously, the novel 2D-UVIF method takes more advantages in enhancement of plasticity of 1060 aluminium alloy.

(a)

(b)

(c)

Figure 6. Processed cylindrical cups. (a) SPIF; (b) LUVIF; (c) 2D-UVIF.”

Comment 2: “Figure 2 should show the moving parts and their trajectories (directions of movement). In the current version the figure does not make any scientific contribution. ” 

Answer: The forming devices and their trajectories are added and displayed in the Fig.1 and Fig.4. And the Fig.2 mainly presents characteristics of the three kinds of incremental forming methods.

Such as”Incremental forming is one kind of variant stamping technology for manufacturing of thin-walled component of metal materials. The rolling blank is cut to be semi-finished specimens with three dimensional size of 80mm×80mm×2mm and 80mm×80mm×1mm on wire electrical discharge machine. The specimens of 1060 sheet and the die are shown in Fig.1(a). Although this forming process is a free-die manufacturing of thin-walled plates, an ordinary die playing an supporting role is facilitated to improvement of forming qualities. The die suffering impact is made of 45 steel with good rigidity and high strength. The combined die is divided into three parts, including the fixture part, the core die part and the blank holder. The core die with replaceable-surface is fixed on the vibrated platform by the fixture part. The specimen is put on the die and compressed by the blank holder and four bolts with preload. Strong pressing force and friction could prevent the free flow of the materials in the flange area of metal sheet. Therefore, the most common phenomenon of wrinkle occurred in the stamping process is avoid. The installation method is shown in Fig.1(b). Because of suffering coupling action of high temperature, violent strike and intense friction, the hemispherical forming tool is made of high temperature alloy of K25 (produced by Baosteel company of China) as shown in Fig.1(c).

(a)

(b)

(c)

Figure 1. Devices for incremental forming: (a) Die and specimen; (b) Installation of die and specimen; (c) Forming tool.” 

The manufacturing experiments of cones and cylindrical cups of 1060 aluminium alloy for traditional incremental forming, longitudinal vibration incremental forming and 2D ultrasonic vibration incremental forming are performed. Firstly, blank sheets (80mm×80mm×2mm) of 1060 aluminium alloy are installed on the cone die and blank sheets (80mm×80mm×1mm) of 1060 aluminium alloy are installed on the cylindrical cup die. Secondly, processing procedure is developed and loaded in the operating system of the specialized incremental forming machine. Thirdly, tool-setting is complete accurately and then the workpiece coordinate system is established. Fourthly, processing parameters are set in the operating system. The hemispherical forming tool feeds horizontally at a constant velocity of 500 mm/min and feeds vertically at a layer depth of 0.1mm as well ass rotating at speed of 500 r/min. Fifthly, the cooling system and the ultrasonic vibration system are opened. Fully synthetic cutting fluid of S318 (produced by Ganis company of China) is poured on the blank sheet of 1060 aluminium alloy and the forming tool of K25 for cooling them down. Besides, the cooling and lubricating fluid is also helpful for improvement of friction conditions and surface qualities. Then, the traditional incremental forming, the longitudinal vibration incremental forming and the 2D ultrasonic vibration incremental forming tests are performed automatically by the processing program. The cone surface and the cylindrical cup surface are split into many contour circle lines by a series of horizontal planes. The forming tool feeds horizontally along the first contour line one round, and then feeds vertically along the inside wall of the die to the second contour line. It constantly recycles the feeding movement until the processing procedure is finished. These continuous trajectories of the kinds of incremental forming methods for cone and cylindrical cup are displayed in Figure 4(a) and (b).

(a)

(b)

Figure 4. The continuous trajectories of the three incremental forming methods: (a) Cone; (b) Cylindrical cup.

Comment 3: “Nothing is specified about the type of trajectory used for SPIF pin processing (contour, spiral, etc). A graphical representation of these trajectories is needed. It is also not specified whether these trajectories are discontinuous (when moving from one Z coordinate to another the tool lifts off the blank) or continuous.” 

Answer: The trajectories of cones and cylindrical cups are shown in Fig.4. They are made of continuous contour lines.

The cone surface and the cylindrical cup surface are split into many contour circle lines by a series of horizontal planes. The forming tool feeds horizontally along the first contour line one round, and then feeds vertically along the inside wall of the die to the second contour line. It constantly recycles the feeding movement until the processing procedure is finished. These continuous trajectories of the kinds of incremental forming methods for cone and cylindrical cup are displayed in Figure 4(a) and (b).

(a)

(b)

Figure 4. The continuous trajectories of the three incremental forming methods: (a) Cone; (b) Cylindrical cup.

In addition, other problems are also corrected. The revised manuscript is seen in the attachment.

We would like to express our great appreciation to you and reviewers for comments on our paper. Looking forward to hearing from you.

Thank you and best regards.

Yours sincerely,

Name: Yuan Lv

  • mail: lvyuan@xust.edu.cn

Mobile Phone: 13572199684

Reviewer 3 Report

Comments and Suggestions for Authors

My Comments are provided in the attached PDF.

Comments on the Quality of English Language

The manuscript is written in quite good English language. There are only some minor mistakes or phrases to be reformulated.

Author Response

Dear Editors and reviewers,

On behalf of my co-authors, we thank you very much for giving us an opportunity to revise our manuscript, we appreciate editors and reviewers very much for their positive and constructive comments and suggestions on our manuscript entitled “Characteristics of 2D Ultrasonic Vibration Incremental Forming of a 1060 Aluminium Alloy Sheet”(ID: Materials-2874144).

We have studied reviewer’s comments carefully and have made revision which marked in red in the paper. We have tried our best to revise our manuscript according to the comments. The revised parts are list as follows.

Comment 1: “The present title ( Effects of 2D Ultrasonic Vibration on Plasticity and Fracture Characteristic of 1060 Aluminium Alloy Sheet for Incremental Forming ) gives emphasis on plasticity and fracture characteristics. But it should be noted that the results are restricted to microhardness (Figure 5.c), residual stresses (Figure 6.c), images of surface morphology (Figure 7) and images of fracture surface of the conical components (Figure 8). In my opinion, the title is beyond the real scope of the manuscript. Please note that in the present version of the manuscript, the authors are always comparing the results obtained by 3 incremental forming methods: the traditional single-point incremental forming (SPIF), the longitudinal ultrasonic vibration incremental forming (LUVIF), and the 2D ultrasonic vibration incremental forming (2D-UVIF). Therefore, the title should be reformulated; and it could become for example:Characteristics of 2D Ultrasonic Vibration compared to other Incremental Forming Methods applied to a 1060 Aluminium Alloy Sheet .” 

Answer: The title has been revised. Characteristics of 2D Ultrasonic Vibration Incremental Forming of a 1060 Aluminium Alloy Sheet”  

Comment 2: “An aspect that the authors should clarify is about the chosen geometry of the manufactured parts. The choice of geometry can be questioned, as cones and parabolas are the easiest (they are less prone to 2 rupture). In terms of formability (and even friction and finishing) I think that the difficulty/challenge is in cylindrical cups.” 

Answer: The manufacturing tests of cylindrical cups and their measurement results are added in this paper.

Such as”2.3.3 Processing Conditions

The manufacturing experiments of cones and cylindrical cups of 1060 aluminium alloy for traditional incremental forming, longitudinal vibration incremental forming and 2D ultrasonic vibration incremental forming are performed. Firstly, blank sheets (80mm×80mm×2mm) of 1060 aluminium alloy are installed on the cone die and blank sheets (80mm×80mm×1mm) of 1060 aluminium alloy are installed on the cylindrical cup die. Secondly, processing procedure is developed and loaded in the operating system of the specialized incremental forming machine. Thirdly, tool-setting is complete accurately and then the workpiece coordinate system is established. Fourthly, processing parameters are set in the operating system. The hemispherical forming tool feeds horizontally at a constant velocity of 500 mm/min and feeds vertically at a layer depth of 0.1mm as well ass rotating at speed of 500 r/min. Fifthly, the cooling system and the ultrasonic vibration system are opened. Fully synthetic cutting fluid of S318 (produced by Ganis company of China) is poured on the blank sheet of 1060 aluminium alloy and the forming tool of K25 for cooling them down. Besides, the cooling and lubricating fluid is also helpful for improvement of friction conditions and surface qualities. Then, the traditional incremental forming, the longitudinal vibration incremental forming and the 2D ultrasonic vibration incremental forming tests are performed automatically by the processing program. The cone surface and the cylindrical cup surface are split into many contour circle lines by a series of horizontal planes. The forming tool feeds horizontally along the first contour line one round, and then feeds vertically along the inside wall of the die to the second contour line. It constantly recycles the feeding movement until the processing procedure is finished. These continuous trajectories of the kinds of incremental forming methods for cone and cylindrical cup are displayed in Figure 4(a) and (b).

(a) (b)

Figure 4. The continuous trajectories of the three incremental forming methods: (a) Cone; (b) Cylindrical cup.

Three cones processed by traditional single point incremental forming, longitudinal ultrasonic vibration incremental forming and 2D ultrasonic vibration incremental forming are shown in figure 5. For its good processing property, these three thick cups of same size without wrinkling and fracture problems are mainly used for micro hardness tests and residual stress tests.

(a)

(b)

(c)

Figure 5. Processed cones. (a) SPIF; (b) LUVIF; (c) 2D-UVIF.

To discover the effect of the 2D ultrasonic vibration mode on the improvement of machined surface qualities and plastic deformation capacity, three 1060 aluminium alloy sheets of 80mm×80mm×1mm are used for manufacturing tests of cylindrical cups of traditional single point incremental forming, longitudinal ultrasonic vibration incremental forming and 2D ultrasonic vibration incremental forming. Because of extensive deformation occurred at the bottom of the side wall, it probably ruptures at this location. Once the rupture phenomenon is detected, processing procedure will be terminated immediately. Their machined surfaces are used for surface topography analysis of laser spectral confocal microscopy (LSCM of KC-H020, produced by Kathmatic company) and micro morphology analysis of scanning electronic microscope (SEM of VEGA, produced by Tescan company of Czech). And their fracture surfaces are probed by SEM technology. These fracture measurements are facilitated to research of micro-structure revolution of 1060 aluminium alloy under coupled conditions of ultrasonic, temperature and stress. Three cylindrical cups of different size are shown in figure 6. And the measurement locations are displayed in Fig.6. The results prove that 2D ultrasonic vibration method is more helpful than longitudinal ultrasonic vibration method for improvement of plastic deformation capacity of 1060 aluminium alloy. Compared to the cylindrical cup of SPIF, the depth of the one of 2D-UVIF is increased by 43.5%. However, the depth of the cylindrical cup of LUVIF is only raised by 17.7%. Obviously, the novel 2D-UVIF method takes more advantages in enhancement of plasticity of 1060 aluminium alloy.

(a)

(b)

(c)

Figure 6. Processed cylindrical cups. (a) SPIF; (b) LUVIF; (c) 2D-UVIF.” 

Comment 3: “Another aspect that the authors should examine is that, in small components, edge effects are more pronounced and can affect deformation. The original specimen used by the authors is a square plate and they are going to make something round, this causes the flap to suffer deformations and nonaxisymmetric slipping. Seems like a bad choice for scientific investigation. And they have no excuse for that as they had a WEDM available (any geometry). Additionally, they didn't leave any details on how they limited the slipping of the sheet metal (With fluting? Knurled? ).” 

Answer: A blank holder is used for incremental forming. The manufacturing tests results proved that there are no wrinkling phenomenons occurred on the cones and cylindrical cups of SPIF, LUVIF and 2D-UVIF. The installation of specimen and die is added in Fig.1.

Such as”Incremental forming is one kind of variant stamping technology for manufacturing of thin-walled component of metal materials. The rolling blank is cut to be semi-finished specimens with three dimensional size of 80mm×80mm×2mm and 80mm×80mm×1mm on wire electrical discharge machine. The specimens of 1060 sheet and the die are shown in Fig.1(a). Although this forming process is a free-die manufacturing of thin-walled plates, an ordinary die playing an supporting role is facilitated to improvement of forming qualities. The die suffering impact is made of 45 steel with good rigidity and high strength. The combined die is divided into three parts, including the fixture part, the core die part and the blank holder. The core die with replaceable-surface is fixed on the vibrated platform by the fixture part. The specimen is put on the die and compressed by the blank holder and four bolts with preload. Strong pressing force and friction could prevent the free flow of the materials in the flange area of metal sheet. Therefore, the most common phenomenon of wrinkle occurred in the stamping process is avoid. The installation method is shown in Fig.1(b). Because of suffering coupling action of high temperature, violent strike and intense friction, the hemispherical forming tool is made of high temperature alloy of K25 (produced by Baosteel company of China) as shown in Fig.1(c).

(a)

(b)

(c)

Figure 1. Devices for incremental forming: (a) Die and specimen; (b) Installation of die and specimen; (c) Forming tool.

Figure 1. Devices for incremental forming: (a) Die and specimen; (b) Installation of die and specimen; (c) Forming tool.” 

Comment 4: “Instead of 3.1. Micro Mechanical Properties it should be 3.1. Microhardness analysis (because there no other mechanical properties).” 

Answer: 3.1 Micro Mechanical Properties is modified into 3.2 Microhardness analysis 

Comment 5: “Readers can question: - Where are located the fracture surfaces of the conical components? Please disclose this issue so that the readers can fully understand what you mean.” 

Answer: Fracture surfaces are located on cylindrical cups not on cones. The manufacturing tests of cylindrical cups and their measurement results are added in this paper. And the SEM locations of fracture surface are displayed in Fig.6.

”To discover the effect of the 2D ultrasonic vibration mode on the improvement of machined surface qualities and plastic deformation capacity, three 1060 aluminium alloy sheets of 80mm×80mm×1mm are used for manufacturing tests of cylindrical cups of traditional single point incremental forming, longitudinal ultrasonic vibration incremental forming and 2D ultrasonic vibration incremental forming. Because of extensive deformation occurred at the bottom of the side wall, it probably ruptures at this location. Once the rupture phenomenon is detected, processing procedure will be terminated immediately. Their machined surfaces are used for surface topography analysis of laser spectral confocal microscopy (LSCM of KC-H020, produced by Kathmatic company) and micro morphology analysis of scanning electronic microscope (SEM of VEGA, produced by Tescan company of Czech). And their fracture surfaces are probed by SEM technology. These fracture measurements are facilitated to research of micro-structure revolution of 1060 aluminium alloy under coupled conditions of ultrasonic, temperature and stress. Three cylindrical cups of different size are shown in figure 6. And the measurement locations are displayed in Fig.6. The results prove that 2D ultrasonic vibration method is more helpful than longitudinal ultrasonic vibration method for improvement of plastic deformation capacity of 1060 aluminium alloy. Compared to the cylindrical cup of SPIF, the depth of the one of 2D-UVIF is increased by 43.5%. However, the depth of the cylindrical cup of LUVIF is only raised by 17.7%. Obviously, the novel 2D-UVIF method takes more advantages in enhancement of plasticity of 1060 aluminium alloy.

(a)

(b)

(c)

Figure 6. Processed cylindrical cups. (a) SPIF; (b) LUVIF; (c) 2D-UVIF.” 

Comment 6: “The manuscript is written in quite good English language. There are some minor mistakes or phrases to be reformulated, as for example:

- Line 58: instead of resent , it should be recent .

- Lines 88 91: The paragraph is too long and hard to be understood. Please avoid long phrases and too many verbs in the same paragraph. (Note the action-verbs: To enhance the volume and surface effects induced by ultrasonic energy and generated on metal materials, a 2D ultrasonic vibration incremental forming method is used to manufacture conical components from 1060 aluminium alloy to determine its plastic deformation behaviours and fracture characteristics in this study ).

- Line 167: instead of 2.3.2 it should be 2.3.3.” 

Answer: The spelling errors have been corrected. And the wrong sub-title is also corrected.

Such as”In recent years, considerable research attentions have been focused on high-energy assisted plastic processing of metal materials.” 

“For enhancement of volume effect and surface effect induced by ultrasonic energy, one kind of 2D ultrasonic vibration incremental forming method is used for manufacturing of cones and cylindrical cups of 1060 aluminium alloy to research on its plastic deformation behaviors and fracture characteristic. The novel 2D ultrasonic vibration incremental forming experiment of cones and cylindrical cups of 1060 aluminium alloy and its contrast experiments are performed in this paper.”

Comment 7: “Line 133: What is decay resistance ? Note how it is: enhance decay resistance ?!” 

Answer: The wrong statement is deleted.

Comment 8: “Section 2. Materials and Methods: In a research article, the experiments section or Materials and Method is an important section. All procedures followed by the authors should be clearly described. It is especially important that readers are informed about all details so that they can repeat the experiments if they are in possession of identical materials and equipment. Details about each relevant piece of equipment used should be provided (namely: version or model, and manufacturer or providers name and country). For example, the i-XRD residual stress analysis system that was used is not revealed. Also, it is relevant to disclose the manufacturers of the 1060 aluminium alloy sheets, and of the hightemperature alloy K2.” 

Answer: All procedures have been clearly described.

Such as”Incremental forming is one kind of variant stamping technology for manufacturing of thin-walled component of metal materials. The rolling blank is cut to be semi-finished specimens with three dimensional size of 80mm×80mm×2mm and 80mm×80mm×1mm on wire electrical discharge machine. The specimens of 1060 sheet and the die are shown in Fig.1(a). Although this forming process is a free-die manufacturing of thin-walled plates, an ordinary die playing an supporting role is facilitated to improvement of forming qualities. The die suffering impact is made of 45 steel with good rigidity and high strength. The combined die is divided into three parts, including the fixture part, the core die part and the blank holder. The core die with replaceable-surface is fixed on the vibrated platform by the fixture part. The specimen is put on the die and compressed by the blank holder and four bolts with preload. Strong pressing force and friction could prevent the free flow of the materials in the flange area of metal sheet. Therefore, the most common phenomenon of wrinkle occurred in the stamping process is avoid. The installation method is shown in Fig.1(b). Because of suffering coupling action of high temperature, violent strike and intense friction, the hemispherical forming tool is made of high temperature alloy of K25 (produced by Baosteel company of China) as shown in Fig.1(c).” 

“As figure 3(b) shown, the 2D ultrasonic vibration auxiliary device (produced by Haituo Machinery Technology company) is fixed on the workbench of a four axis CNC machining center (produced by Jinan First Machine Tool company). A fixture and a conical-surface die are installed on the vibrated platform of 2D ultrasonic device. Besides, a hemispherical tooling is designed for replacing the traditional milling tooling. Then, the universal milling machine is transformed to be a specialized 2D ultrasonic vibration incremental forming machine. ”

“Fully synthetic cutting fluid of S318 (produced by Ganis company of China) is poured on the blank sheet of 1060 aluminium alloy and the forming tool of K25 for cooling them down. ”

“Once the rupture phenomenon is detected, processing procedure will be terminated immediately. Their machined surfaces are used for surface topography analysis of laser spectral confocal microscopy (LSCM of KC-H020, produced by Kathmatic company) and micro morphology analysis of scanning electronic microscope (SEM of VEGA, produced by Tescan company of Czech).”

“In order to probe the distribution and magnitude of residual stress field on processed surface of cones for traditional single point incremental forming, longitudinal ultrasonic vibration incremental forming and 2D ultrasonic vibration incremental forming, three sets of residual stress tests are performed on i-XRD residual stress analysis system (μ-X360, produced by Pulstec industrial company of Japan). The measurement conditions are list in table 5.”

Comment 9: “There is no justification for the values of the operating/processing parameters chosen in the milling machine or in the ultrasound. This aspect will also be questionable by the readers.” 

Answer: Processing parameters of the milling machine and ultrasonic device are added.

 Such as” The signal generator launches a ultrasonic wave signal that is amplified by the amplifier. Then, the strengthening signal drives the horn to vibrate at frequency of 20kHz and amplitude of 10μm.” 

“The manufacturing experiments of cones and cylindrical cups of 1060 aluminium alloy for traditional incremental forming, longitudinal vibration incremental forming and 2D ultrasonic vibration incremental forming are performed. Firstly, blank sheets (80mm×80mm×2mm) of 1060 aluminium alloy are installed on the cone die and blank sheets (80mm×80mm×1mm) of 1060 aluminium alloy are installed on the cylindrical cup die. Secondly, processing procedure is developed and loaded in the operating system of the specialized incremental forming machine. Thirdly, tool-setting is complete accurately and then the workpiece coordinate system is established. Fourthly, processing parameters are set in the operating system. The hemispherical forming tool feeds horizontally at a constant velocity of 500 mm/min and feeds vertically at a layer depth of 0.1mm as well ass rotating at speed of 500 r/min. Fifthly, the cooling system and the ultrasonic vibration system are opened. Fully synthetic cutting fluid of S318 (produced by Ganis company of China) is poured on the blank sheet of 1060 aluminium alloy and the forming tool of K25 for cooling them down. Besides, the cooling and lubricating fluid is also helpful for improvement of friction conditions and surface qualities. Then, the traditional incremental forming, the longitudinal vibration incremental forming and the 2D ultrasonic vibration incremental forming tests are performed automatically by the processing program. The cone surface and the cylindrical cup surface are split into many contour circle lines by a series of horizontal planes. The forming tool feeds horizontally along the first contour line one round, and then feeds vertically along the inside wall of the die to the second contour line. It constantly recycles the feeding movement until the processing procedure is finished. These continuous trajectories of the kinds of incremental forming methods for cone and cylindrical cup are displayed in Figure 4(a) and (b).”

Comment 10: “Under the conditions of the milling machine, we will have forced friction between the rotation of the tool and the translation of the tool on the sheet. Do the authors have left the tool free to rotate depending on the tangential speed in the contact area? As it is, it seems that they are ruining the surface finish..” 

Answer: In the processing procedures, the rotating tool is free. Cooling fluid is used for decrease the friction.

“Fifthly, the cooling system and the ultrasonic vibration system are opened. Fully synthetic cutting fluid of S318 (produced by Ganis company of China) is poured on the blank sheet of 1060 aluminium alloy and the forming tool of K25 for cooling them down. Besides, the cooling and lubricating fluid is also helpful for improvement of friction conditions and surface qualities.”

Comment 11: “The SPIF process always requires oil or a sacrificial polymeric sheet to avoid scratching the surface of the part. I didn't see this information on the manuscript. The AA1060 "sticks" to the tool. What is the surface finish of the tool?” 

Answer: ”Fifthly, the cooling system and the ultrasonic vibration system are opened. Fully synthetic cutting fluid of S318 (produced by Ganis company of China) is poured on the blank sheet of 1060 aluminium alloy and the forming tool of K25 for cooling them down. Besides, the cooling and lubricating fluid is also helpful for improvement of friction conditions and surface qualities.” 

Comment 12: “In fact, the article does not include any measurements of the surface finish. This is mandatory for work of this type. In manufacturing it is the first thing you look at..” 

Answer: The introduction part has been rewrite. (1) Research status about the traditional single-point incremental forming and the one-way ultrasonic incremental forming are given. Critical conclusion about the shortage of current research is described.

Such as”3.1. Surface topography Analysis

To investigate the characteristics of machined surface finish of cylindrical cups processed by different forming technologies, a set of surface topography tests are conducted. Sampling locations are displayed in Fig.6. Then, three surface topography images are shown in Fig.7. As the figures shown, red color represents peak height and blue color represents valley depth. It is found that there are similar range of maximum peak height and maximum valley depth in the three images. But totally different topography features are distributed on the machined surfaces of the cylindrical cups.

(a)

(b)

(c)

Figure 7. Machined Surface topography images of cylindrical cups. (a) SPIF; (b) LUVIF; (c) 2D-UVIF.

To quantitatively analyze topography characteristics and roughness of machined surface of the three cylindrical cups processed by SPIF, LUVIF and 2D-UVIF methods, the measurement results of surface topography parameters, including mean roughness (Ra), maximum peak height (Rp), root mean square deviation (Rq), maximum valley depth (Rv) and maximum height difference between maximum peak height and maximum valley depth (Rz), are present in Table 2-Table 4. As the three tables shown, mean roughness of machined surface of LUVIF cylindrical cup is larger than the other two ones. 2D-UVIF technology produces minimum mean roughness of machined surface of cylindrical cup. Moreover, parameters of Rp, Rq, Rv and Rz feature the same laws. Obviously, the novel 2D-UVIF method takes good advantages in machined surface qualities.

Table 2. Measurement results of surface topography parameters of SPIF cylindrical cups (Units: μm).

Topography Parameters

Max Value

Min Value

Mean Value

Standard Deviation

Mean Roughness/Ra

3.150

1.691

2.421

1.032

Maximum Peak Height/Rp

9.679

5.425

7.552

3.008

Root Mean Square Deviation/Rq

3.657

2.097

2.877

1.103

Maximum Valley Depth/Rv

-5.136

-7.459

-6.298

1.643

Maximum Height Difference/Rz

17.138

10.562

13.850

4.650

Table 3. Measurement results of surface topography parameters of LUVIF cylindrical cups (Units: μm).

Topography Parameters

Max Value

Min Value

Mean Value

Standard Deviation

Mean Roughness/Ra

3.077

1.888

2.661

0.670

Maximum Peak Height/Rp

10.693

5.544

8.594

2.703

Root Mean Square Deviation/Rq

4.031

2.311

3.346

0.912

Maximum Valley Depth/Rv

-5.617

-8.708

-7.590

1.713

Maximum Height Difference/Rz

19.401

11.162

16.184

4.406

Table 4. Measurement results of surface topography parameters of 2D-UVIF cylindrical cups (Units: μm).

Topography Parameters

Max Value

Min Value

Mean Value

Standard Deviation

Mean Roughness/Ra

2.081

1.794

1.938

0.203

Maximum Peak Height/Rp

6.622

5.851

6.237

0.545

Root Mean Square Deviation/Rq

2.474

2.260

2.367

0.151

Maximum Valley Depth/Rv

-4.993

-7.004

-5.999

1.422

Maximum Height Difference/Rz

12.855

11.615

12.235

0.877

Comment 13: “Results of microhardness shown in Figure 5(c) are a bit puzzling. The values are relatively high. Annealed AA1060 should be on the order of HV22. HV of 100 seems exaggerated to me. Have the surfaces been properly prepared for hardness measurement?.” 

Answer: Micro-hardness values of unprocessed surface of raw material are in range of 37-42HV. Micro-hardness values of LUVIF part are up to 100. Hardening effect induced by longitudinal ultrasonic vibration is the main reason.

Comment 14: “All figures should be legible. Terminology used in the figures should be in accordance with the terminology used in the text. For example, the following inconsistencies appear:

- In lines 104 and 113 it is mentioned mould , but in Figure 1(a) it is mentioned die , as well as in Figure 3(b).

- In line 143 it is mentioned vibration platform, but in Figure 3(b) it is mentioned Vibrator Platform ..” 

Answer: All the wrong expressions have been rewrite.

Such as”Incremental forming is one kind of variant stamping technology for manufacturing of thin-walled component of metal materials. The rolling blank is cut to be semi-finished specimens with three dimensional size of 80mm×80mm×2mm and 80mm×80mm×1mm on wire electrical discharge machine. The specimens of 1060 sheet and the die are shown in Fig.1(a). Although this forming process is a free-die manufacturing of thin-walled plates, an ordinary die playing an supporting role is facilitated to improvement of forming qualities. The die suffering impact is made of 45 steel with good rigidity and high strength. The combined die is divided into three parts, including the fixture part, the core die part and the blank holder. The core die with replaceable-surface is fixed on the vibrated platform by the fixture part. The specimen is put on the die and compressed by the blank holder and four bolts with preload. Strong pressing force and friction could prevent the free flow of the materials in the flange area of metal sheet. Therefore, the most common phenomenon of wrinkle occurred in the stamping process is avoid. The installation method is shown in Fig.1(b). Because of suffering coupling action of high temperature, violent strike and intense friction, the hemispherical forming tool is made of high temperature alloy of K25 (produced by Baosteel company of China) as shown in Fig.1(c).

(a)

(b)

(c)

Figure 1. Devices for incremental forming: (a) Die and specimen; (b) Installation of die and specimen; (c) Forming tool.” 

Comment 15: “For sake of comparison, dimensions (in mm) of the obtained conical parts (depicted in Figure 4) should be clearly mentioned..” 

Answer: Dimensions of processed cones and cylindrical cups are marked.

Such as”Three cones processed by traditional single point incremental forming, longitudinal ultrasonic vibration incremental forming and 2D ultrasonic vibration incremental forming are shown in figure 5. For its good processing property, these three thick cups of same size without wrinkling and fracture problems are mainly used for micro hardness tests and residual stress tests.

(a)

(b)

(c)

Figure 5. Processed cones. (a) SPIF; (b) LUVIF; (c) 2D-UVIF.

To discover the effect of the 2D ultrasonic vibration mode on the improvement of machined surface qualities and plastic deformation capacity, three 1060 aluminium alloy sheets of 80mm×80mm×1mm are used for manufacturing tests of cylindrical cups of traditional single point incremental forming, longitudinal ultrasonic vibration incremental forming and 2D ultrasonic vibration incremental forming. Because of extensive deformation occurred at the bottom of the side wall, it probably ruptures at this location. Once the rupture phenomenon is detected, processing procedure will be terminated immediately. Their machined surfaces are used for surface topography analysis of laser spectral confocal microscopy (LSCM of KC-H020, produced by Kathmatic company) and micro morphology analysis of scanning electronic microscope (SEM of VEGA, produced by Tescan company of Czech). And their fracture surfaces are probed by SEM technology. These fracture measurements are facilitated to research of micro-structure revolution of 1060 aluminium alloy under coupled conditions of ultrasonic, temperature and stress. Three cylindrical cups of different size are shown in figure 6. And the measurement locations are displayed in Fig.6. The results prove that 2D ultrasonic vibration method is more helpful than longitudinal ultrasonic vibration method for improvement of plastic deformation capacity of 1060 aluminium alloy. Compared to the cylindrical cup of SPIF, the depth of the one of 2D-UVIF is increased by 43.5%. However, the depth of the cylindrical cup of LUVIF is only raised by 17.7%. Obviously, the novel 2D-UVIF method takes more advantages in enhancement of plasticity of 1060 aluminium alloy.

(a)

(b)

(c)

Figure 6. Processed cylindrical cups. (a) SPIF; (b) LUVIF; (c) 2D-UVIF.” 

Comment 16: “In Figure 6(c), write MPa instead of Mpa ; and write Sampling Location Number instead of simply Sampling Number . Additionally, note that error bars are probably missing in the results shown in the plot/graph.” 

Answer: “Unit: MPa” and “ Sampling Location Number” are rewrite. Error bars in the figure are added.

Comment 17: “In Figures 7 and 8, a ruler (in m) is shown for each photo. Therefore, it does not make sense to write the magnifications, i.e., please delete 500×,1000×,2000×, etc..” 

Answer: magnifications of “500×,1000×,2000×  are deleted.

”Figure 10. Magnification images of surface morphology of 1060 aluminium alloy sheet, SPIF part, LUVIF part and 2D-UVIF part. (a), (b), (c) 1060 sheet; (d), (e) ,(f) SPIF; (g), (h), (i) LUVIF; (j), (k), (l) 2D-UVIF.” 

Figure 11. Micro-morphology images of fracture surface of the cylindrical cups for SPIF, LUVIF and 2D-UVIF. (a), (b), (c) SPIF; (d), (e), (f), LUVIF; (g), (h), (i), 2D-UVIF.

Comment 18: “The readers will question about the exact location of each the observations shown in Figures 7 and 8..” 

Answer: Locations of surface morphology of cylindrical cups and micro-structure of fracture surface are marked in the figure 6.

”To discover the effect of the 2D ultrasonic vibration mode on the improvement of machined surface qualities and plastic deformation capacity, three 1060 aluminium alloy sheets of 80mm×80mm×1mm are used for manufacturing tests of cylindrical cups of traditional single point incremental forming, longitudinal ultrasonic vibration incremental forming and 2D ultrasonic vibration incremental forming. Because of extensive deformation occurred at the bottom of the side wall, it probably ruptures at this location. Once the rupture phenomenon is detected, processing procedure will be terminated immediately. Their machined surfaces are used for surface topography analysis of laser spectral confocal microscopy (LSCM of KC-H020, produced by Kathmatic company) and micro morphology analysis of scanning electronic microscope (SEM of VEGA, produced by Tescan company of Czech). And their fracture surfaces are probed by SEM technology. These fracture measurements are facilitated to research of micro-structure revolution of 1060 aluminium alloy under coupled conditions of ultrasonic, temperature and stress. Three cylindrical cups of different size are shown in figure 6. And the measurement locations are displayed in Fig.6. The results prove that 2D ultrasonic vibration method is more helpful than longitudinal ultrasonic vibration method for improvement of plastic deformation capacity of 1060 aluminium alloy. Compared to the cylindrical cup of SPIF, the depth of the one of 2D-UVIF is increased by 43.5%. However, the depth of the cylindrical cup of LUVIF is only raised by 17.7%. Obviously, the novel 2D-UVIF method takes more advantages in enhancement of plasticity of 1060 aluminium alloy.

(a)

(b)

(c)

Figure 6. Processed cylindrical cups. (a) SPIF; (b) LUVIF; (c) 2D-UVIF.” 

In addition, other problems are also corrected. The revised manuscript is seen in the attachment.

We would like to express our great appreciation to you and reviewers for comments on our paper. Looking forward to hearing from you.

Thank you and best regards.

Yours sincerely,

Name: Yuan Lv

  • mail: lvyuan@xust.edu.cn

Mobile Phone: 13572199684

Round 2

Reviewer 2 Report

Comments and Suggestions for Authors

The authors have provided acceptable solutions to all the issues raised by the reviewer. Therefore, I consider the paper suitable for publication in the journal.

Reviewer 3 Report

Comments and Suggestions for Authors

The authors have seriously considered all my detailed comments (from Comment 1 till Comment 18). They have changed the title; they now included the manufacture of cylindrical cups; made considerable modifications; added 3 new figures, and 2 new tables; etc. Compared to the previous version, this new version is much better and, since the previously indicated flaws have been repaired, in my opinion the present manuscript fulfils now the requirements for being published.

Comments on the Quality of English Language

The revised version deserves now to be published after some minor editing of English language.